# Differential Roles of Five Fluffy Genes (*flbA*–*flbE*) in the Lifecycle In Vitro and In Vivo of the Insect–Pathogenic Fungus *Beauveria bassiana*

**DOI:** 10.3390/jof8040334

**Published:** 2022-03-23

**Authors:** Chong-Tao Guo, Xin-Cheng Luo, Sheng-Hua Ying, Ming-Guang Feng

**Affiliations:** Institute of Microbiology, College of Life Sciences, Zhejiang University, Hangzhou 310058, China; 11707030@zju.edu.cn (C.-T.G.); 12207006@zju.edu.cn (X.-C.L.); yingsh@zju.edu.cn (S.-H.Y.)

**Keywords:** entomopathogenic fungi, upstream developmental activators, gene expression and regulation, aerial conidiation, blastospore production, virulence, stress response

## Abstract

The fluffy genes *flbA*–*flbE* are well-known players in the upstream developmental activation pathway that activates the key gene *brlA* of central developmental pathway (CDP) to initiate conidiation in *Aspergillus nidulans*. Here, we report insignificant roles of their orthologs in radial growth of *Beauveria bassiana* under normal culture conditions and different stresses although *flbA* and *flbD* were involved in respective responses to heat shock and H_2_O_2_. Aerial conidiation level was lowered in the deletion mutants of *flbB* and *flbE* (~15%) less than of *flbA* and *flbC* (~30%), in which the key CDP genes *brlA* and *abaA* were repressed consistently during normal incubation. The CDP-controlled blastospore production in submerged cultures mimicking insect hemolymph was abolished in the *flbA* mutant with *brlA* and *abaA* being sharply repressed*,* and decreased by 55% in the *flbC* mutant with only *abaA* being downregulated. The fungal virulence against a model insect was attenuated in the absence of *flbA* more than of *flbC* irrespective of normal cuticle infection or cuticle-bypassing infection (intrahemocoel injection). These findings unravel more important role of *flbA* than of *flbC*, but null roles of *flbB/D/E*, in *B. bassiana*’s insect–pathogenic lifecycle and a scenario distinctive from that in *A**.nidulans*.

## 1. Introduction

The fluffy genes *flbA*–*flbE* encode signal transducers that are well-known players in the upstream developmental activation (UDA) pathway to activate the key activator gene *brlA* of central developmental pathway (CDP) required for initiation of conidiation in *Aspergillus nidulans*, as well reviewed [1,2,3]. The signal transducers are the G-protein signaling protein FlbA, the b-ZIP transcription factor (TF) FlbB, the TF FlbC with two C_2_H_2_ zinc finger DNA-binding domains, the c-Myb TF FlbD and the FlbB-interacting protein FlbE, respectively. The expression of *brlA* is activated through the cascades *fluG-flbA*, *fluG-flbC* and *fluG-**flbE/flbB/flbD**,* in which *fluG* acts as a core UDA regulator [4,5,6,7,8,9,10,11,12,13,14]. The activated *brlA* leads to sequential activation of the downstream CDP genes *abaA* and *wetA* and the velvety family gene *vosA*, which are collectively essential for conidial production and maturation [15,16,17,18]. The fluffy genes were found in the early analyses of repressive fluffy mutations, which led to functional loss of *brlA* and ‘fluffy’ (conidiation abolished) colony morphology [19,20,21,22]. Such fluffy genes have corresponding orthologs in many filamentous fungal genomes annotated in the past decade. The post-genomic era has witnessed increasing evidences of ascomycetous genome divergence, making it necessary to reconsider whether the genetic control principles on the asexual development of *A. nidulans* are applicable to Pezizomycotina [3,23,24,25]. The necessity is emphasized by the existence of an approximately one-fold molecular difference between small [437–534 amino acids (aa)] and large (860–914 aa) FluG homologs and of multiple FluG-like regulators similar to small FluG in different lineages of ascomycetes [13,26]. Even in aspergilli, *fluG* may not necessarily play the same regulatory role in asexual development as elucidated in *A. nidulans*. For example, conidial yield of *fluG* null mutant was moderately decreased in *A. flavus* [27] but not affected at all in *A. niger* [28].

The insect–pathogenic fungus *Beauveria bassiana* (Hypocreales: Cordycipitaceae) has the broadest host spectrum among all insect pathogens [29] and serves as a main source of fungal pesticides to combat against wide-spectrum arthropod pests [30]. Designing or improving large-scale production technology of high-quality conidia as active ingredients of fungal pesticides requires the knowledge of regulatory mechanisms underlying asexual developmental activation. In *B. bassiana*, three CDP genes (*brlA*, *abaA* and *wetA*) and downstream *vosA* regulate conidial production and maturation [31,32] as documented in *A. nidulans* [1,2,3], although the two fungi have distinctive conidiation modes, namely the formation of spore balls (clustered conidia) on tiny zigzag rachises (conidiophores) and of chained conidia on phialides. Previously, disruption of *fluG* in *B. bassiana* led to a very limited defect (~10% decrease) in aerial conidiation but a sharp increase in submerged blastospore production, accompanied by time-course-active transcription profiles of all CDP genes and *flbA*–*flbE* in both aerial and submerged cultures [26]. As the coding gene of regulator of G-protein signaling, *Bbrgs1* orthologous to *flbA* in *A. nidulans* was reported to mediate conidiation and heat tolerance in *B. bassiana* but play no role in the fungal virulence [33]. Similar to aerial conidiation, blastospore production of *B. bassiana* is an asexual developmental process under the control of either *brlA* or *abaA* and crucial for the fungal proliferation by yeast-like budding in insect hemocoel to accelerate insect death from mummification [32]. Although null role of *fluG* in the fungal UDA pathway was shown in our previous study [26], it remains unknown whether and how most of those fluffy genes act as regulators of asexual developmental processes in *B. bassiana*. In this study, we generated and analyzed single-gene deletion and complementation mutants of *flbA*–*flbE* in order to elucidate their functions in the insect–pathogenic lifecycle of *B. bassiana*. An emphasis was placed upon a possible role of each target gene in conidiation, blastospore production in vitro andin vivo, and a possibility of its transcriptional link to *brlA* or *abaA*.

## 2. Materials and Methods

### 2.1. Bioinformatic Analysis of Fungal FlbA–FlbE Orthologs

The amino acid sequences of *A. nidulans* FlbA–FlbE (NCBI accession numbers: EAA58402, CBF79600, EAA64532, EAA66152 and EAA65198, respectively) were used as queries to search through the NCBI genome databases of *B. bassiana* [34] and some other ascomycetous fungi including entomopathogens and non-entomopathogens via BLASTp analysis (http://blast.ncbi.nlm.nih.gov/blast.cgi, accessed on 20 March 2022). Conserved domains and nuclear localization signal (NLS) motif were predicted from each query protein and its *B. bassiana* ortholog at http://smart.embl-heidelberg.de/ (accessed on 20 March 2022) and http://nls-mapper.iab.keio.ac.jp/ (accessed on 20 March 2022), respectively. The sequence identities of each *B. bassiana*Flb protein to those orthologues found in other fungal species were analyzed by sequence alignment with a program at http://www.bio-soft.net/format/DNAMAN.htm/ (accessed on 20 March 2022), followed by phylogenetic analysis with a maximum likelihood method in MEGA7 at http://www.megasoftware.net/ (accessed on 20 March 2022).

### 2.2. Subcellular Localization of FlbA–FlbE in B. bassiana

Green fluorescence-tagged fusion proteins of FlbA–FlbE were expressed in the wild-type strain *B. bassiana* ARSEF 2860 (designated WT) as described previously for expression of FluG-GFP fusion protein [26]. Briefly, the coding sequence of each target gene was amplified from the WT cDNA with paired primers (Appendix A) and ligated to the N-terminus of *gfp* (GenBank accession U55763) in the vector pAN52-C-gfp-bar using a one-step cloning kit (Vazyme, Nanjin, China). In the vector, capital C denotes the cassette 5′-*Pme*I-*Spe*I-*Eco*RV-*Eco*RI-*Bam*HI-3′ driven by the homologous promoter P*tef1* [35,36]. The resultant vector pAN52- *x*-gfp-bar (*x* = *flbA*, *flbB*, *flbC*, *flbD* or *flbE*) was integrated into the WT strain via *Agrobacterium*-mediated transformation. Putative transgenic strains were screened by the *bar* resistance to phosphinothricin (200 μg/mL). For each transformation, a transgenic strain showing strong green fluorescence signal was incubated for full conidiation on Sabouraud dextrose agar (4% glucose, 1% peptone and 1.5% agar) plus 1% yeast extract (SDAY). Conidia from the culture were suspended in SDBY (i.e., agar-free SDAY) and incubated at optimal 25 °C for 3 d in the light/dark (L:D) cycles of 0:24, 12:12 and 24:0 on a shaking bed (150 rpm). Hyphal samples from the cultures were stained with DAPI (4′,6′-diamidine-2′-phenylindole dihydrochloride; Sigma-Aldrich, Shanghai, China) and visualized with laser scanning confocal microscopy (LSCM). The green fluorescence intensity of a fixed circular area of cytoplasm or nucleus was quantified from each of 33*–*35 cells in the hyphae of the culture grown in each L:D cycle by means of the software ImageJ at https://imagej.nih.gov/ij/ (accessed on 20 March 2022) and used to compute the ratio of nuclear versus cytoplasmic green fluorescence intensity (N/C-GFI) as an index of relative accumulation level of each fusion protein in the nucleus of each cell.

### 2.3. Generation of Targeted Gene Mutants

The disruption strategy of *fluG* in our previous study [26] was adopted to generate null mutants of *f**lbA**–**f**lbE* by deleting full-length coding and partial flanking DNA sequences of each target gene from the WT genome through homologous recombination of its 5′ and 3′ flanking fragments separated by *bar* marker in the vector p0380-5′*x*-bar-3′*x* (Appendix A). The constructed vectors were individually integrated into the WT strain as aforementioned. Further, the full-length coding sequence of each target gene with flanking regions was amplified from the WT DNA and ligated to the *Hin*dIII/*Xba*I sites in p0380-sur-gateway to exchange for the gateway fragment. The resultant p0380-sur-*x* was ectopically integrated into an identified Δ*flb* mutant for targeted gene complementation in the same transformation system. Putative mutants were screened by the *bar* resistance to phosphinothricin (200 μg/mL) or the *sur* resistance to chlorimuron ethyl (10 μg/mL). Expected recombinant events in the colonies were identified through PCR (Appendix A) and real-time quantitative PCR (qPCR) analyses (Appendix A). Listed in Appendix A are pairs of primers used for amplification of DNA fragments and detection of targeted DNA and cDNA samples. The identified deletion mutant (DM) of each gene and its complementation mutant (CM) were evaluated in parallel with the parental WT strain in the following experiments of three independent replicates per strain unless specified otherwise.

### 2.4. Assays for Growth Rates under Normal Culture Conditions and Stresses

For all DM and control (WT and CM) strains, 1μL aliquots of a 10^6^ conidia/mL suspension were spotted on the plates of rich medium SDAY, 1/4 SDAY (amended with 1/4 of each SDAY nutrient), minimal medium Czapek–Dox agar (CDA; 3% sucrose, 0.3% NaNO_3_, 0.1% K_2_HPO_4_, 0.05% KCl, 0.05% MgSO_4_ and 0.001% FeSO_4_ plus 1.5% agar) and CDAs amended with different carbon or nitrogen sources. After a 7-day incubation at the optimal regime of 25 °C in a light/dark (L:D) cycle of 12:12, the diameter of each colony was measured as a growth index using two measurements taken perpendicular to each other across the center. Typical colony images were also collected.

The same method was also used to initiate colony growth on CDA plates alone (control) or supplemented with oxidants, osmotic agents and cell wall perturbing agents as described previously [26]. Cellular response to heat shock was assayed by exposing normal 2 d-old SDAY colonies to 42 °C for 3 and 6 h, followed by 5-d growth recovery at 25 °C. Colony images and diameter values were collected as aforementioned. Relative growth inhibition (RGI) of each DM or control strain under a given stress was computed as an index of its sensitivity to the stress using the formula RGI = (*d*_c_–*d*_s_)/*d*_c_×100, where *d*_c_ and *d*_s_ denote the respective diameters of control and stressed colonies. For the Δ*flbD* mutant more sensitive to H_2_O_2_ than its control strains, gradient H_2_O_2_ concentrations (1.0*–*3.5 mM) were added to CDA plates for comparison and verification of their responses to the oxidative stress induced by H_2_O_2_.

### 2.5. Assays for Conidial Yield and Quality

The cultures for quantification of biomass and conidiation capacity were initiated by spreading 100 μL aliquots of a 10^7^ conidia/mL suspension on SDAY plates (9 cm diameter) overlaid with or without cellophane and incubated for 9 d at the optimal regime of 25 °C and L:D 12:12. During the period of incubation, three samples were taken from each cellophane-free plate culture on days 5, 7 and 9 using a cork borer (5 mm diameter). Conidia in each sample were released into 1 mL of 0.02% Tween 80 via a 10-min supersonic vibration, followed by assessing the conidial concentration in the suspension with a hemocytometer and converting it to the number of conidia per square centimeter of plate culture. Biomass levels were assessed from three cellophane-overlaid SDAY cultures on days 3, 5 and 7, respectively. The conidial quality of each strain was assessed as the indices of median germination time (GT_50_, h) at 25 °C, median lethal time (LT_50_, min) for tolerance to a 45 °C wet–heat stress and median lethal dose (LD_50_, J/cm^2^) for resistance to UVB irradiation (weighted wavelength: 312 nm), as described elsewhere [26].

### 2.6. Insect Bioassays

The fifth-instar larvae of greater wax moth (*Galleria mellonella*) were assayed for the virulence of each DM or control strain in two infection modes. Briefly, normal cuticle infection (NCI) was initiated by immersing a group of ~35 larvae (three groups per strain) for 10 s in 40 mL of a 10^7^ conidia/mL suspension. Cuticle-bypassing infection (CBI) was initiated by injecting 5 μL of a 10^5^ conidia/mL suspension into the hemocoel of each larva in each of three groups. All groups of larvae inoculated for NCI or CBI were maintained at 25 °C. Their survival/mortality records were taken every 12 h (CBI) or 24 h (NCI). The time-mortality trend in each group was subjected to modeling analysis for the estimation of LT_50_ (d) as a virulence index via NCI or CBI.

### 2.7. Analyses of Virulence-Related Cellular Events

For the deletion mutants significantly compromised in virulence, several cellular events essential for NCI and hemocoel colonization were examined or analyzed in parallel with their control strains. To reveal the impact of a deleted gene on initiation of NCI, conidial adherence to insect cuticle was assessed on locust (*Locusta migratoria manilensis*) hind wings pre-treated in 37% H_2_O_2_ as described elsewhere [37]. Every 5 μL of a 10^7^ conidia/mL suspension in sterile water was spotted on the center of each hind wing attached to 0.7% water agar, followed by an 8-h incubation at 25 °C. Counts of conidia were made immediately from three microscopic fields of each wing and repeated after 30 s washing in sterile water. The percent ratio of post-wash versus pre-wash counts was computed as an index of conidia adherence to the wing cuticle with respect to the WT strain. Due to a reliance of conidial adherence upon hydrophobicity [38,39,40,41,42], conidial hydrophobicity of each strain was assessed in an aqueous-organic system as described previously [43,44]. After hyphal invasion into insect body, hemocoel colonization relies upon the yeast-like budding proliferation of hyphal bodies (i.e., blastospores) formed by the hyphae through dimorphic transition under the control of the key CDP genes *brlA* and *abaA* [32] and is tightly linked to a speed of host death from mummification [26,41]. To reveal a status of proliferation in vivo, the abundance of hyphal bodies was microscopically examined in the hemolymph samples taken from surviving larvae 96 h post-NCI or 72 h post-CBI. The hemocytometer was used to assess the concentration of hyphal bodies from each of three samples per larva (three larvae per strain) taken 72–216 h post-NCI or 60–108 h post-CBI as described previously [26]. Further, 100 mL aliquots of a 10^6^ conidia/mL suspension in trehalose-peptone broth (TPB), a medium amended from CDB (i.e., agar-free CDA) with 3% trehalose as sole carbon source and 0.3% peptone as sole nitrogen source to mimic insect hemolymph, were incubated for 5 d on the shaking bed at 25 °C. From day 2 onwards, blastospore concentration and biomass level (mg/mL) were measured daily from each culture to estimate dimorphic transition rate (no. blastospores/mg biomass) in vitro as a reference to the proliferation in vivo.

### 2.8. Transcriptional Profiling

The qPCR analysis was performed to verify expected recombinant events in the mutants and gain insight into their phenotypic changes. Briefly, cellophane-overlaid SDAY and submerged TPB cultures were initiated as aforementioned and incubated for 7 and 5 d at the optimal regime, respectively. From the end of a 48-h incubation onwards, total RNA was extracted daily from each of the SDAY or TPB cultures under the action of RNAiso Plus Kit (TaKaRa, Dalian, China), and reversely transcribed into cDNA under the action of PrimeScript RT reagent kit (TaKaRa). The cDNA samples derived from three independent cultures on each sampling occasion were used as templates in qPCR analysis to assess: (1) daily transcript levels of *flbA**–flbE* in the WT cultures grown for 2–7 d on SDAY; (2) transcript levels of each *flb* gene in the 3 d-old SDAY and TPB cultures of its DM and control strains; (3) daily transcript levels of the key CDP genes *brlA* and *abaA* in the SDAY and TPB cultures of each DM compromised in asexual development and its control strains; and (4) transcript levels of 38 phenotype-related genes in the 3 d-old SDAY cultures of each DM compromised in a given phenotype and its control strains. The analyzed genes are well known in function, including the coding genes of five superoxide dismutases (Sod1–Sod5), six catalases (Cat1–Cat6), five hydrophobin or hydrophobin-like proteins (Hyd1–Hyd5), 11 subtilisin-like Pr1 family proteases and 11 heat-shock family proteins. The qPCR analysis with paired primers (Appendix A) was performed using the SYBR Premix *ExTaq* kit (TaKaRa). The transcript level of the fungal β-actin gene was used as an internal standard. A threshold-cycle (2^−^^ΔΔ^^Ct^) method was used to compute relative transcript levels for: (1) each *flb* gene in the daily SDAY cultures of the WT strain with respect to the standard at the end of 48 h incubation; (2) each *flb* gene in the 3 d-old SDAY or TPB cultures of related mutants with respect to the WT standard; (3) *brlA* and *abaA* in the daily SDAY and TPB cultures of related mutants with respect to the WT standard; and (4) phenotype-related genes in the 3 d-old SDAY cultures of related mutants with respect to the WT standard. One-fold transcript change was considered as a significant level of down- or upregulation for each of the analyzed genes.

### 2.9. Statistical Analysis

All experimental data were subjected to one-way analysis of variance and Tukey’s honestly significant difference (HSD) test for phenotypic differences between each DM and its control strains.

## 3. Results

### 3.1. Phylogenetic Linkages and Sequence Comparison of Fungal FlbA-FlbE Orthologs

BLASTp search with the queries of *A. nidulans* resulted in identification of FlbA, FlbB, FlbC, FlbD and FlbE orthologs in *B. bassiana* and selected ascomycetous fungi. The orthologs of each Flb protein were found in most, but not all, of the surveyed fungal genomes, and clustered in phylogeny to distinctive clades or subclades obviously associated with fungal lineages (Appendix A). FlbA (EJP68072), FlbB (EJP63983), FlbC (EJP70334), FlbD (EJP63935) and FlbE (EJP69751) in *B. bassiana* shared higher sequence identities with their orthologs in Hypocreales than in other orders. The ortholog of FlbD in *Cordyceps militaris* was exceptionally clustered to an orphan clade distant from the clades of its orthologs in Cordyciptaceae and other fungal lineages and shared a sequence identity of only 39% with the *B. bassiana* FlbD.

Conserved domain analysis revealed a structural similarity of each Flb protein between *B. bassiana* and *A. nidulans* (Figure 1A). FlbA features a C-terminal RGS (Regulator of G protein signaling) domain and two DEP domains that have been proposed to play a selective role in targeting DEP domain-containing proteins to specific subcellular membranous sites [45,46]. A BRLZ (basic region leucin zipper) domain usually present in eukaryotic bZIP domain-containing transcription factors [47] was predicted from the FlbB sequence of *B. bassiana* at an e-value of 0.0558 but not predictable from the counterpart of *A. nidulans*, as shown previously [48]. FlbC and FlbD have two C-terminal ZnF_C_2_H_2_ domains and two N-terminal SANT (SWI3, ADA2, N-CoR and TFIIIB) domains involved in DNA binding [49], respectively. The SANT domains of FlbD were also revealed in a previous analysis [3]. However, no distinguishable domain was predicted from the FlbE sequence of *B. bassiana* or *A. nidulans**,* although previous sequence analysis of FlbE revealed a linkage of multiple regions (domains) with subcellular localization of FlbB in *A. nidulans* [14]. In addition, an NLS motif was predicted from the amino acid sequences of all Flb proteins in the two fungi but not predictable from the *A. nidulans* FlbE sequence.

### 3.2. Transcription Profiles and Subcellular Localization of FlbA–FlbE in B. bassiana

During a period of 7-d incubation on SDAY at the optimal regime, five *flb* genes showed differential transcription profiles in the WT cultures with respect to the standard of each at the end of 48 h incubation (Figure 1B). The expression of *flbB* was repressed by more than 80% consistently during the period while *flbE* was downregulated significantly on most sampling occasions. In contrast, *flbA* and *flbC* were increasingly upregulated or expressed at high levels during the period, followed by transcript level of *flbD* fluctuating around the standard. These data demonstrated that, in the WT strain, *flbA* and *flbC* were far more active than *flbD* at transcriptional level while *flbB* and *flbE* were repressively expressed during the period of normal incubation

The nuclear localization of each Flb protein suggested by the predicted NLS motif was confirmed by the expression of green fluorescence-tagged fusion proteins in the WT strain. As shown in LSCM images (Figure 1C), the fusion proteins accumulated more in the nuclei than in the cytoplasm of hyphal cells stained with DAPI (shown in red) regardless of the hyphae from the cultures grown in an L:D cycle of 0:24, 12:12 or 24:0. The N/C-GFI ratios assessed in the hyphal cells from the three L:D cycles were, on average, 1.92, 2.16 and 2.04 for FlbA-GFP, 2.08, 2.20 and 2.04 for FlbB-GFP, 2.89, 3.04 and 3.25 for FlbC-GFP, and 2.29, 2.08 and 2.18 for FlbD-GFP, respectively (Figure 1D). These observations revealed that FlbC accumulated in the nuclei significantly more than did FlbA, FlbB and FlbD (Tukey’s HSD, *p* < 0.01) and that each Flb protein accumulated more in the nuclei than in the cytoplasm in a fashion independent of light. However, we failed to express FlbE-GFP in the WT strain in many attempts, leaving its subcellular localization unexplored.

### 3.3. Differential Roles of flbA–flbE in Radial Growth, Aerial Conidiation and Stress Tolerance

The deletion mutants of five *flb* genes grew as well as their control strains during a 7-d incubation after initiation of colony growth by spotting 10^3^ conidia on the plates of rich medium SDAY, 1/4 SDAY and minimal medium CDA at the optimal regime. The colonies of all mutants did not show any fluffy phenotype (Figure 2A), and were similar to those of control strains in diameter (Figure 2B). As an exception, the Δ*flbA* mutant showed a moderate increase in colony size on CDA. Similar colony growth also occurred on CDAs amended with different carbon (glucose, trehalose, fructose, lactose, maltose, mannitol, glycerol, sodium acetate, olive oil and oleic acid) or nitrogen (NaNO_2_, NH_4_Cl and NH_4_NO_3_) sources (Appendix A). In the assays for cellular responses to stress cues, only the Δ*flbA* and Δ*flbD* mutants were significantly more sensitive than their control strains to a 3- or 6-h heat shock at 42 °C during normal growth and an oxidative stress induced with 2 mM H_2_O_2_, respectively (Figure 2C). Both of them exhibited null responses to the other oxidant menadione, three osmotic agents (NaCl, KCl and sorbitol) and two cell wall perturbing agents (Congo red and calcofluor white) as did the remaining deletion mutants (Appendix A). The elevation of the Δ*flbD* mutant’s sensitivity to H_2_O_2_ was further clarified by its responses to gradient concentrations of H_2_O_2_ (Figure 2D) and its growth inhibition percentage increased from 11% at 1.0 mM to 28% at 3.0 mM (Figure 2E) in comparison to the responses of its control strains.

Next, SDAY cultures were initiated by spreading 100 μL aliquots of a 10^7^ conidia/mL suspension to quantify conidial yields and accumulated biomass levels. During a 9-d incubation at the optimal regime, the conidial yields of Δ*flbA*, Δ*flbB* and Δ*flbC* were reduced by 75%, 65% and 65% on day 5, respectively, in comparison to the mean WT yield of 22.1 × 10^7^ conidia/cm^2^ plate culture (Figure 3A). The yield reductions diminished to 62%, 43% and 31% on day 7 and to 33%, 30% and 14% on day 9, at the time of which the WT yield reached 66.0 × 10^7^ conidia/cm^2^ plate culture. The Δ*flbE* mutant exhibited a significant yield decrease of ~15% on day 7 or 9. The accumulated biomass levels in the cellophane-overlaid SDAY cultures were increased by 28%, 24% and 43% in Δ*flbB* on days 3, 5 and 7, and 21% and 38% in Δ*flbE* on days 3 and 7, but decreased by 31% and 42% in Δ*flbA* and 33% and 32% in Δ*flbC* on days 5 and 7, respectively (Figure 3B). Neither conidial yield nor biomass level was affected in the Δ*flbD* cultures. Among the indices of conidial quality, moreover, GT_50_ as a viability index was moderately increased by 12–15% in Δ*flbA*, Δ*flbB* and Δ*flbE* compared to the WT strain (Figure 3C), followed by heat tolerance decreased by 18% in Δ*flbA* (Figure 3D) and UVB resistance lowered by ~20% in both Δ*flbB* and Δ*flbE* (Figure 3E).

All mentioned phenotypes were well restored to the WT levels by targeted gene complementation. The experimental data demonstrated dispensable roles for all of five *flb* genes in the radial growth of *B. bassiana* under normal culture conditions but significant roles for *flbA* and *flbD* in the fungal responses to heat shock and H_2_O_2_, respectively. The reduced conidial yields of the Δ*flbA* and Δ*flbC* mutants correlated with their lowered biomass accumulation levels. Such correlation was not seen in the Δ*flbB* and Δ*flbE* mutants, which were facilitated in biomass accumulation but less compromised in conidiation. The fungal conidiation capacity was eventually reduced by ~30% in the absence of *flbA* or *flbC*, ~15% in the absence of *flbB* or *flbE*, but not affected in the absence of *flbD*. These results indicated that fluffy phenotype was not caused by the deletion of each *flb* gene in *B. bassiana* as was documented in *A. nidulans* [1,2].

### 3.4. Differential Roles of flbA–flbE in Host Infection and Virulence-Related Cellular Events

NCI and CBI in the standardized bioassays resulted in the mean LT_50_ values of 5.05 and 3.49 d for the WT strain against *G. mellonella* larvae, respectively. Compared to these mean values, the LT_50_s of Δ*flbA* and Δ*flbC* were prolonged significantly (Tukey’s HSD, *p*< 0.05) by 45% and 17% via NCI (Figure 4A) and 11% and 6% via CBI (Figure 4B), respectively. The Δ*flb**B* LT_50_ was slightly prolonged via CBI but not via NCI. The remaining deletion mutants showed little virulence change via either infection mode in comparison to their control strains.

For insight into significantly attenuated virulence of Δ*flbA* and Δ*flbC* via both NCI and CBI, cellular events critical for NCI and hemocoel colonization by proliferation in vivo were compared between the two mutants and their control strains. As a crucial trait for initiation of NCI, conidial adherence to locust wing cuticle was lowered by 29% and 23% for Δ*flbA* and Δ*flbC* relative to the WT strain, respectively (Figure 4C), accompanied by 52% and 33% reductions in conidial hydrophobicity (Figure 4D) determinant to the adherence [38,39,40,41,42]. A linear correlation was highly significant between the measurements of conidial hydrophobicity and adherence from the tested strains (r^2^ = 0.975, *F*_1,19_ = 116.7, *p* < 0.0017).

Next, a status of yeast-like budding proliferation in insect hemocoel to accelerate host death from mummification was examined under a microscope. Hyphal bodies formed by the WT strain were far more abundant than those formed by the Δ*flbA* mutant in the hemolymph samples taken from the larvae surviving 96 h post-NCI or 72 h post-CBI (Figure 4E). Consequently, the concentrations of hyphal bodies formed by the WT strain in the samples on average were 1.5 × 10^6^, 3.3 × 10^6^, 5.3 × 10^6^ and 5.9 × 10^6^ cells/mL at 72, 96, 120 and 144 h post-NCI (Figure 4F), and 2.0 × 10^6^, 5.3 × 10^6^, 8.9 × 10^6^ and 10.8 × 10^6^ cells/mL at 60, 72, 84 and 96 h post-CBI (Figure 4G), respectively. In contrast, the corresponding Δ*flbA* concentrations after NCI and CBI were not measurable on the first sampling occasion, and decreased, respectively, by 98% and 96%, 86% and 64%, and 81% and 43% on the following sampling occasions. The concentrations of hyphal bodies produced by the Δ*flbC* mutant after NCI and CBI decreased by 45% and 20% on the first sampling occasion, and the reductions diminished to only 5% and 11% on the last sampling occasion, respectively. These data demonstrated that the formation and proliferation in vivo of hyphal bodies were blocked in the absence of *flbA* much more than of *flbC* and highlighted more important role of *flbA* than of *flbC* in the adaptation of *B. bassiana* insect-pathogenic lifestyle.

Submerged blastospore production in vitro serves as a reference to the status of dimorphic (hypha-blastospore) essential for the fungal proliferation in vivo. In the 3 d-old TPB cultures mimicking insect hemolymph, biomass level was markedly enhanced by 55% in Δ*flbA* among the tested mutants and control strains (Figure 5A). Intriguingly, blastospore production was abolished in Δ*flbA* and reduced by 55% in Δ*flbC* (Figure 5B). Microscopic examination of culture samples revealed no blastospore formation in the Δ*flbA* cultures (Figure 5C). Further time-course monitoring of the Δ*flbA* cultures demonstrated a biomass accumulation level increasingly enhanced by 54–146% (Figure 5D) and a dimorphic transition rate consistently abolished (Figure 5E) during the period of 2- to 5-d incubation. Despite insignificant changes in biomass production during the period, dimorphic transition rate in the Δ*flbC* cultures decreased by 49% on day 2, 43% on day 5, but increased significantly by 26% on day 5 in comparison to the measurements from the WT cultures. These results uncovered a true fluffy phenotype in the submerged TPB cultures of Δ*flbA* but different changes of blastospore production not associated with the biomass levels in the Δ*flbC* cultures.

In *B. bassiana*, both aerial conidiation and submerged blastospore production in vitro are asexual developmental processes genetically controlled by the key CDP gene *brlA* or *abaA* [32]. In the present study, the expression of *brlA* in the TPB cultures of the Δ*flbA* mutant versus the WT strain was repressed by 94–99% (nearly abolished) during the period of a 5-d incubation (Figure 5E), accompanied by sharp repression of *abaA* by 66–96% during the same period (Figure 5F). In contrast, the time-course transcript levels of *abaA* alone were reduced by 86–97% in the TBP cultures of the Δ*flbC* mutant.

The above results indicated more important role of *flbA* than of *flbC,* but dispensable roles of *flbB*, *flbD* and *flbE*, in *B. bassiana*’s host infection and hemocoel colonization. The more attenuated virulence of Δ*flbA* than of Δ*flbC* via either NCI or CBI was due to two reasons. Compared to *flbC*, *flbA* was more involved in the conidial hydrophobicity and adherence required for initiation of NCI. Second, *flbA* was more involved in the transcriptional activation of both *brlA* and *abaA* to mediate dimorphic transition, which is essential for the fungal blastospore production in vitro and proliferation in vivo by yeast-like budding to speed up host mummification to death [32].

### 3.5. Linkages of Altered Phenotypes with Transcriptional Changes of RelatedGenes

Transcript levels of some functionally characterized genes were assessed by qPCR analysis to gain an in-depth insight into the phenotypic defects of those Δ*flb* mutants. Among the analyzed genes, five of six catalase genes were downregulated by 55–85% in the Δ*flbD* mutant relative to the WT strain (Figure 6A), including *cat2/catB* and *cat5/catP* as major contributors to total catalase activity required for decomposition of H_2_O_2_ in *B. bassiana* [50]. Expression levels of 11 heat shock protein genes [51,52,53] were all sharply repressed or even abolished in Δ*flbA* (Figure 6B). The repressed genes correlated well with an elevated sensitivity of Δ*flbD* to H_2_O_2_ and of Δ*flbA* to a 3- or 6-h heat shock at 42 °C during the normal growth at 25 °C.

As key CDP activator genes, *brlA* and *abaA* were significantly downregulated in the SDAY cultures of Δ*flbA* on days 3–7 and 2–6, respectively, during the period of a 7-d incubation at the optimal regime (Figure 6C). The two CDP genes were also downregulated in the Δ*flbC* cultures in most samples, but differentially expressed in the remaining Δ*flb* mutants at insignificant levels with respect to the WT standard. The transcript changes of *brlA* and *abaA* correlated with the conidiation defects observed in Δ*flbA* and Δ*flbC* but were not indicative of less suppressed conidiation in the Δ*flbB* and Δ*flbE* mutants.

Five hydrophobin genes (*hyd1*–*hyd5*) and 11 Pr1 protease genes were analyzed to reveal their impacts on the NCI process of the Δ*flbA* and Δ*flbC* mutants compromised in virulence. Among those, *hyd1* and *hyd2* are reported to mediate the biosynthesis of classes I and II hydrophobins and their assembly into an outermost rodlet-bundle layer of conidial coat determinant to conidial hydrophobicity and adherence to insect cuticle [38]. In this study, *hyd1* and *hyd2* were more downregulated in the 3 d-old SDAY cultures of Δ*flbA* (80% and 87%) than of Δ*flbC* (65% and 59%) relative to the WT strain, accompanied by two of three other function-unknown *hyd* genes differentially expressed in Δ*flbA* but not affected in Δ*flbC* (Figure 6D). Previously, five Pr1 genes (*pr1A2*, *pr1B1*, *pr1B2*, *pr1C* and *pr1G*) were confirmed as significant contributors to total activity of secreted Pr1 proteases and success of NCI while the remaining Pr1 genes were redundant in function [54]. In the present study, three of the five functional Pr1 genes were markedly downregulated in the 3 d-old SDAY cultures of Δ*flbA* (*pr1A2*, *pr1B1* and *pr1G*) and Δ*flbC* (*pr1B2*, *pr1C* and *pr1G*), accompanied by four and three other Pr1 genes significantly repressed in the two mutants, respectively (Figure 6E). The transcript changes of these genes were well restored by targeted gene complementation, giving an explanation for blocked NCI and attenuated virulence in the absence of *flbA* or *flbC*.

## 4. Discussion

As presented above, the protein sequences encoded by five *flb* genes in *B. bassiana* were similar to the corresponding orthologs in *A. nidulans* and four of them were shown to accumulate more in nucleus than in cytoplasm irrespective of incubation under light or in full darkness. Their deletion mutants showed no fluffy phenotype in normal plate cultures. This is different from fluffy phenotypes caused by the loss-of-function mutations of *flbA****–****flbE* in *A. nidulans* [1,2]. Indeed, our Δ*flb* mutants were not compromised in radial growth on rich and scant media under normal culture conditions and also under different types of stresses. Exceptionally, the Δ*flbA* growth was facilitated moderately on CDA and CDAs amended with some of tested carbon or nitrogen sources but suppressed significantly by heat shock. The defects of Δ*flbA* and Δ*flbC* in aerial conidiation were more conspicuous than those of Δ*flbB* and Δ*flbE* but mitigated to limited levels with increasing incubation time. The true fluffy phenotype was observed only in the submerged Δ*flbA* cultures mimicking insect hemolymph, resulting in blocked proliferation in vivo and reduced virulence. The Δ*flbD* mutant displayed insignificant changes in all examined phenotypes except its increased sensitivity to oxidative stress induced by H_2_O_2_ and its involvement in the expression of most catalase genes. The time-course transcription profiles of *brlA* and *abaA* in the SDAY cultures were correlated with the conidiation defects in Δ*flbA* and Δ*flbC* but not with those in Δ*flbB* and Δ*flbE*. These results uncover that five *flb* genes function in *B. bassiana* in a fashion distinctive from their orthologs in *A. nidulans*, as discussed below.

Our data demonstrated substantial roles of *flbA* and *flbC* in the early activation of key CDP genes to initiate aerial conidiation in *B. bassiana* [32] due to consistently repressed expression of either *brlA* or *abaA* in their plate cultures during normal incubation. Particularly, the role of *flbA* in asexual development was further clarified by abolished blastospore production and nearly abolished expression of both *brlA* and *abaA* in the submerged Δ*flbA* cultures. This is well in accordance with the in vitro blastospore production abolished in the absence of *brlA* or *abaA* [32] and suggests a transcriptional link of *flbA* to *brlA* in *B. bassiana* as elucidated previously in *A. nidulan*s [4,20]. The transcriptional repression of *abaA* instead of *brlA* in the submerged Δ*flbC* cultures led to the blastospore production significantly reduced at the early stage of incubation but increased at the late stage, hinting at closer link of *flbC* to *abaA* than to *brlA* at transcriptional level. However, it remains elusive how *flbA* or *flbC* acts as an activator of *brlA* or *abaA* in the present study because *fluG* has been shown to play no role in the UDA pathway of *B. bassiana* [26] and hence differ from the regulatory role of its homolog characterized in the same pathway of *A. nidulans* [1,2,3]. We tried to explore a possible role for *flbA* or *flbC* in activating the expression of *brlA* or *abaA* through yeast one-hybrid assays, but failed in repeated attempts because strong automatic activities of the promoters P*flbA*, P*flbC*, P*brlA* and P*abaA* ligated to the plasmid pAbAi led to an infeasibility to perform the yeast assays. Recently, AbaA was evidently bound to the promoter of the velvet protein gene *veA* in *Metarhizium robertsii* [55], suggesting a distinctive linkage between *abaA* and *veA* in the fungal insect pathogen.

Aside from greater role in asexual development, *flbA* was more important than *flbC* for the adaptation of *B. bassiana* to insect–pathogenic lifestyle and was involved in transcriptional mediation of multiple genes encoding heat-shock proteins. The importance is well presented by more attenuated virulence of Δ*flbA* than of Δ*flbC* via NCI or CBI. The virulence difference between the two mutants was likely due to differential repression of their key *hyd* genes, which determine conidial hydrophobicity and adherence required for initiation of NCI [37,38,39]. The difference also could be attributable to differentially delayed or blocked proliferation in vivo by yeast-like budding, which determines a speed of host mummification to death [26,40,41,42,56]. Importantly, the blocked proliferation in vivo of Δ*flbA* was evidenced by the abolition of blastospore production in vitro and the drastic repression of both *brlA* and *abaA* in the TPB cultures mimicking insect hemolymph. This highlights an essentiality of *flbA* for colonization of host hemocoel by *B. bassiana*. Notably, our Δ*flbA* mutant was much less compromised in aerial conidiation than was the deletion mutant of *Bbrgs1*
*(**flbA*) constructed previously in the background of another *B. bassiana* strain (ARSEF 252) and also different from the previous mutant not compromised in virulence despite their similar defects in heat tolerance [33]. We speculate that the differential phenotypes of the two Δ*flbA* mutants could be largely attributed to a big (>10-fold) difference in conidiation capacity between the two WT strains. In our WT strain, opposite rhythms of two frequency (FRQ) proteins (Frq1 and Frq2) in nucleus can persistently activate the expression of CDP genes in a circadian day to orchestrate nonrhythmic conidiation for a high yield of ~55 × 10^7^ conidia/cm^2^ achieved in 7 or 8 d-old plate cultures or on insect cadaver surfaces regardless of photoperiod change [57,58]. However, most genomes of several *B. bassiana*s trains available in the NCBI databases contain a single FRQ rather than two FRQs, which exist only in the WT strain used in the present study.

In conclusion, our study unravels transcriptional links of *flbA* and *flbC* to *brlA* and/or *abaA* in *B. bassiana* and greater role of *flbA* than of *flbC* in fungal conidiation, blastospore production and insect–pathogenic lifecycle. However, *flbB*, *flbD* and *flbE* were not influential on the expression of *brlA* or *abaA* irrespective of limited or little contribution to conidiation. Neither were they involved in submerged blastospore production and cellular events associated with host infection, hemocoel colonization and virulence. In addition, *flbA* and *flbD* were involved in the fungal responses to heat shock and H_2_O_2_, respectively, and also in the expression of related stress-responsive genes. These findings offer a novel insight into the roles of five ‘fluffy’ genes that are distinctive from those characterized in *A. nidulans*.

## Figures and Tables

**Figure 1 jof-08-00334-f001:**
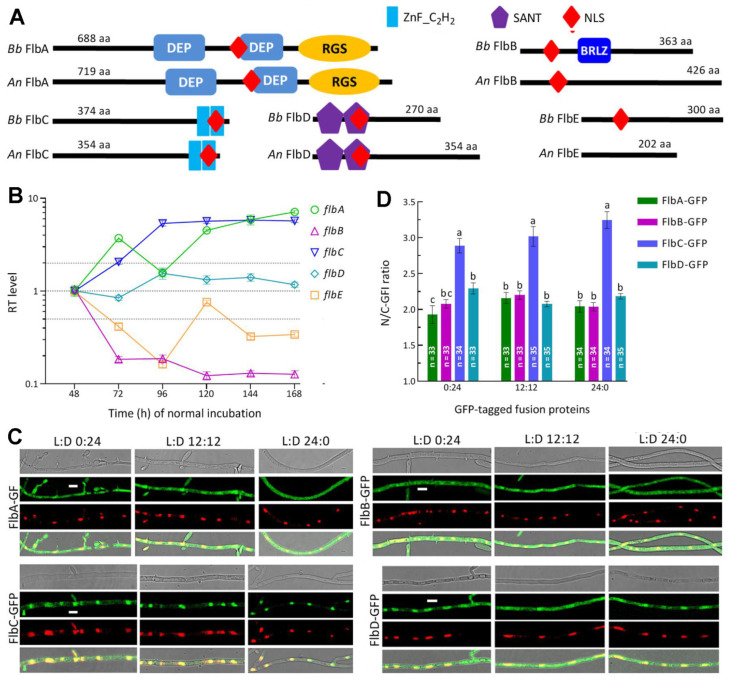
Sequence, transcription and subcellular features of FlbA–FlbE in *B. bassiana*. (**A**) Sequence comparison of FlbA–FlbE between *B. bassiana* (*Bb*) and *A. nidulans* (*An*). Domains and nuclear localization signal (NLS) were predicted from each protein sequence at http://smart.embl-heidelberg.de/ (accessed on 20 March 2022) and http://nls-mapper.iab.keio.ac.jp/ (accessed on 20 March 2022), respectively. (**B**) Relative transcript (RT) levels of *f**lbA–f**lbE* in the SDAY cultures of wild-type *Bb* strain (WT) during a 7-d incubation at the optimal regime of 25 °C in a light/dark (L:D) cycle of 12:12 with respect to the standard level on day 2. (**C**) LSCM images (scales: 5 μm) for subcellular localization of green fluorescence-tagged Flb fusion proteins expressed in the WT strain. Cell samples were taken from the 3 d-old SDBY cultures grown at 25 °C in the respective L:D cycles of 0:24, 12:12 and 24:0 and stained with the nuclear dye DAPI (shown in red). Each four-image panel show bright, expressed (green), stained (red nuclei) and overlapped (yellow nuclei) images of the same field, respectively. (**D**) Nuclear versus cytoplasmic green fluorescence intensity (N/C-GFI) ratios of the fusion proteins in the hyphal cells. Error bars denote standard deviations (SDs) of the means from three cDNA samples analyzed via qPCR (**B**) or 33 to 35 cells in the examined hyphae (**D**).

**Figure 2 jof-08-00334-f002:**
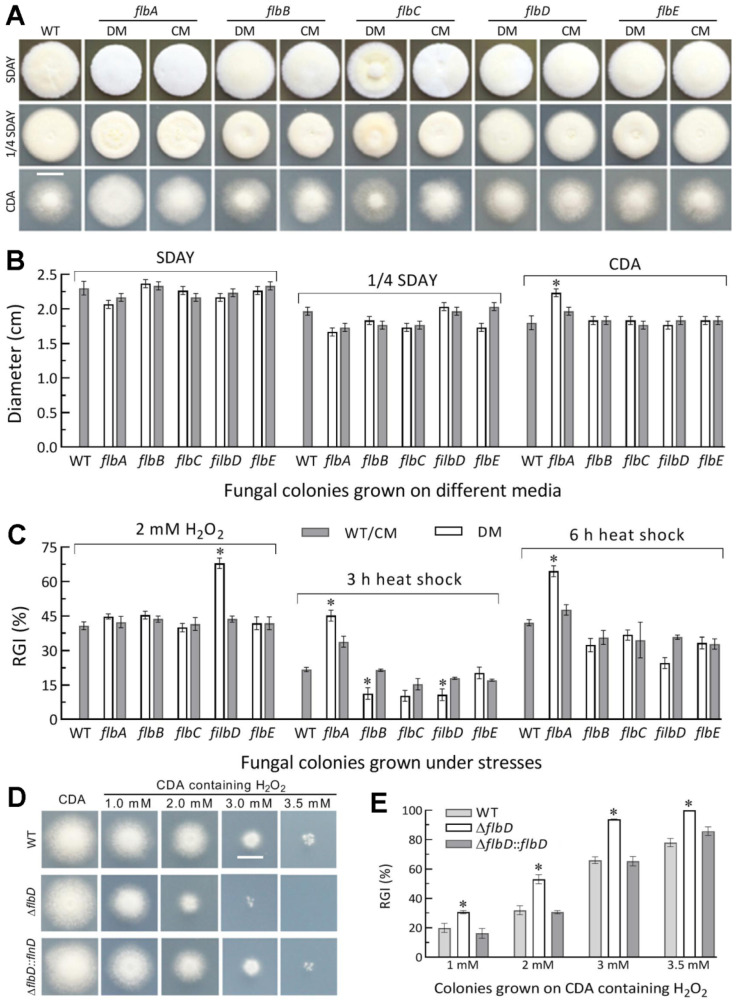
Radial growth rates of Δ*flb* mutants (DM) and control (WT and CM) strains in *B. bassiana*. (**A**,**B**) Images (scale: 10 mm) and diameters of fungal colonies grown at the optimal regime of 25 °C and L:D 12:12 for 7 d on rich medium SDAY, 1/4 SDAY and minimal medium CDA after initiated with 10^3^ conidia. (**C**) Relative growth inhibition (RGI) percentages of fungal colonies incubated at 25 °C for 7 d on CDA containing H_2_O_2_ and of SDAY colonies incubated at 25 °C for 5-d growth recovery after exposing 2 d-old colonies to a 42 °C heat shock for 3 or 6 h. (**D**,**E**) Images (scale: 10 mm) and relative growth inhibition of the Δ*flbD* and control strains incubated for 6.5 d on H_2_O_2_-containing CDA plates. * *p* < 0.05 (Tukey’s HSD). Error bars: SDs from three replicates.

**Figure 3 jof-08-00334-f003:**
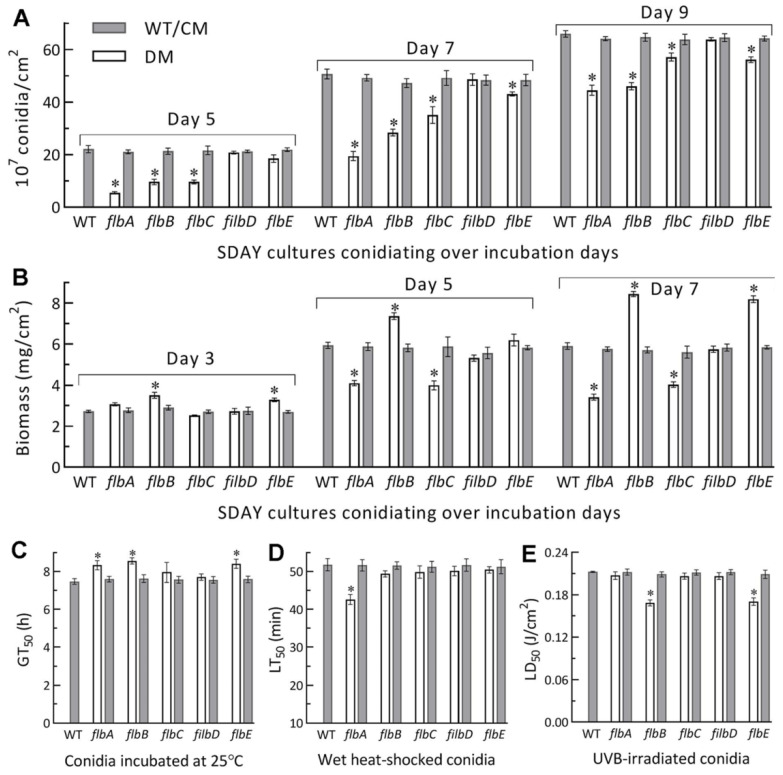
Aerial conidiation and conidial quality of Δ*flb* mutants (DM) and their control (WT and CM) strains in *B. bassiana*. (**A**,**B**) Conidial yields and biomass levels measured from the SDAY cultures during a 9-d incubation at the optimal regime of 25 °C and L:D 12:12, respectively. The cultures were initiated by spreading 100 μL aliquots of a 10^7^ conidia/mL suspension. (**C**–**E**) Median germination time GT_50_ (h) for conidial viability at 25 °C, LT_50_ (min) for conidial tolerance to a 45 °C wet–heat stress and LD_50_ (J/cm^2^) for conidial resistance to UVB irradiation, respectively. **p*< 0.05 in Tukey’s HSD tests. Error bars: SDs of the means from three independent replicates.

**Figure 4 jof-08-00334-f004:**
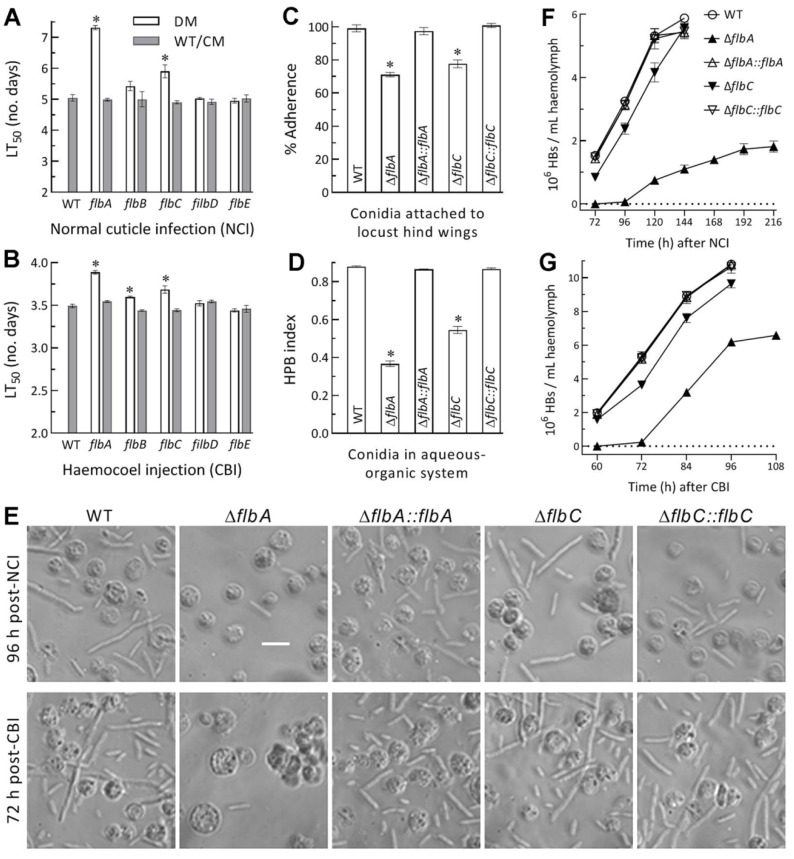
Virulence and related cellular events of Δ*flb* mutants (DM) and control (WT and CM) strains in *B. bassiana*. (**A**,**B**) LT_50_s (d) estimated by fitting the time-mortality trends of *G.*
*mellonella* larvae after topical application (immersion) of a 10^7^ conidia/mL suspension for normal cuticle infection (NCI) and intrahemocoel injection of ~500 conidia per larva for cuticle-bypassing infection (CBI), respectively. (**C**) Conidial adherence to locust hind wing cuticle assessed as percent ratios of post-wash versus pre-wash counts with respect to the WT standard. (**D**) Conidial hydrophobicity (HPB) index quantified in an aqueous-organic system. (**E**) Microscopic images (scale bar: 20 μm) for the status of hyphal bodies (HBs; slender cells) and host hemocytes (spherical or subspherical cells) in the hemolymph samples taken from surviving larvae 120 h post-NCI and 72 h post-CBI. (**F**,**G**) Concentrations of hyphal bodies in the hemolymph samples taken over the time after NCI and CBI, respectively. * *p* < 0.05 in Tukey’s HSD tests. Error bars: SDs of the means from three independent replicates.

**Figure 5 jof-08-00334-f005:**
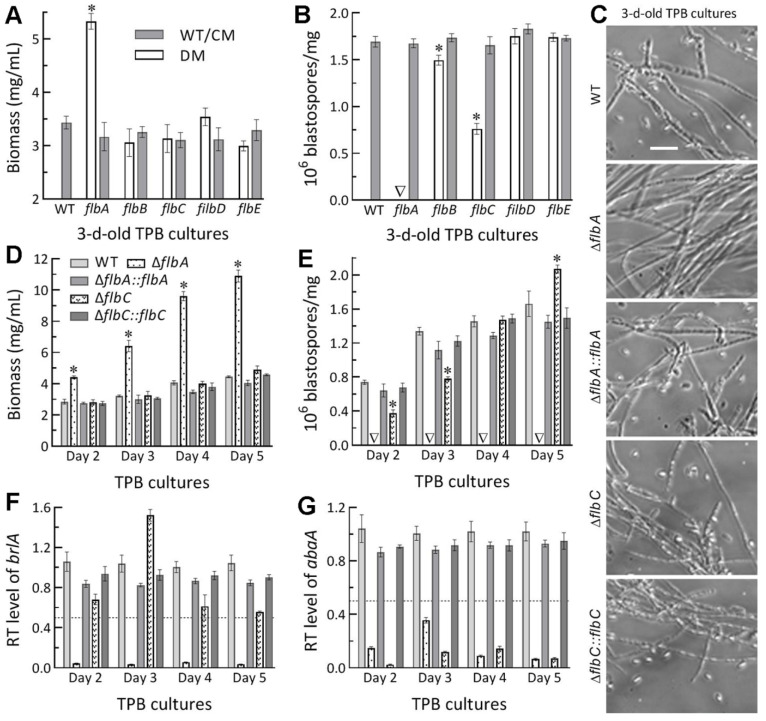
Submerged blastospore production of Δ*flb* mutants (DM) and their control (WT and CM) strains in *B. bassiana*. (**A**,**B**) Biomass levels and dimorphic transition rates measured from the 3 d-old TPB cultures grown at 25 °C, respectively. (**C**) Microscopic images (scale: 20 μm) for a status of blastospore production in the 3 d-old cultures of a 10^6^ conidia/mL TPB mimicking insect hemolymph. (**D**,**E**) Time-course biomass levels and dimorphic transition rates in the TPB cultures incubated at 25 °C for 2–5 d on a shaking bed. (**F**,**G**) Relative transcript (RT) levels of the key CDP genes *brlA* and *abaA* in the TPB cultures of mutants with respect to the WT standard during the 5-d incubation. The dashed line denotes a significant level of one-fold (50%) downregulation. * *p* < 0.05 (**A**,**B**,**D**,**E**) in Tukey’s HSD tests. Error bars: SDs of the means from three independent replicates.

**Figure 6 jof-08-00334-f006:**
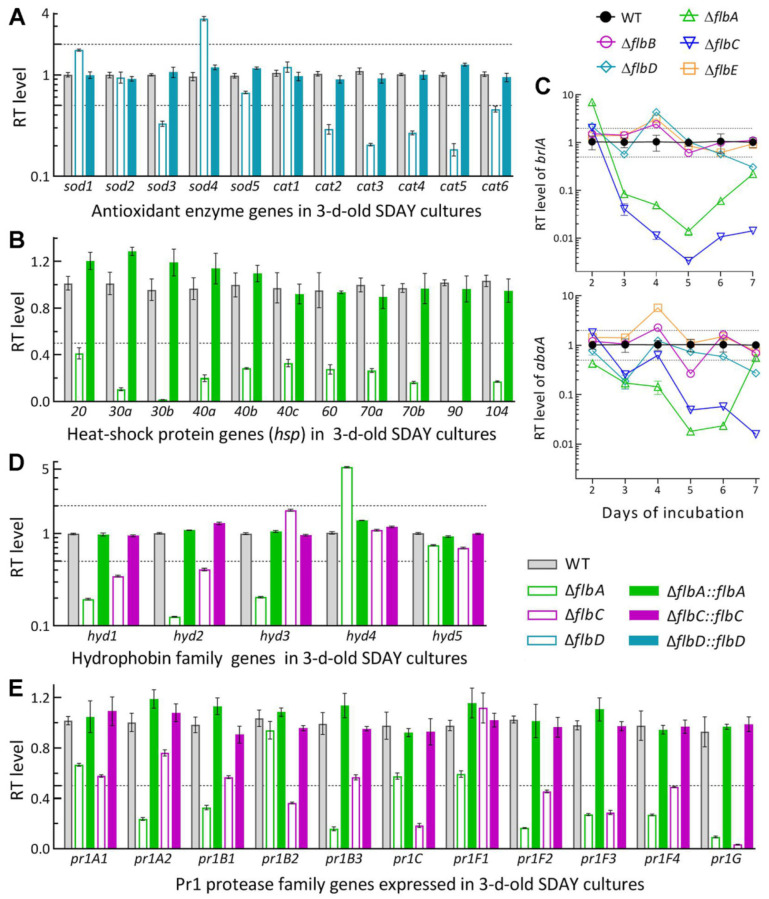
Relative transcript (RT) levels of clustered genes associated with phenotypes of the *flb* mutants with respect to the WT standard in *B. bassiana*. The tested cDNA samples were derived from the SDAY cultures initiated by spreading 100 μL aliquots of a 10^7^ conidia/mL suspension and incubated for 3 d (**A**,**B**,**D**,**E**) or 2–7 d (**C**) at the optimal regime of 25 °C and L:D 12:12. The lower and/or upper dashed lines denote significant levels of one-fold down- and upregulation, respectively. Error bars: SDs of the means from three independent cDNA samples examined by qPCR analysis.

## Data Availability

All data presented in this study are included in the paper and associated Appendix A.

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
