# Peer review of "Differential Roles of Five Fluffy Genes (flbAflbE) in the Lifecycle In Vitro and In Vivo of the Insect–Pathogenic Fungus Beauveria bassiana"

_jof, 2022, doi:10.3390/jof8040334_

Round 1

Reviewer 1 Report

In this work, Guo and colleagues generate single gene deletion mutants of the B. bassiana orthologs of A. nidulans UDA genes flbA, flbB, flbC, flbD and flbE, which in this model Eurotiomycete are essential for the induction of asexual development. The authors assess the effect of single gene deletions on asexual cell production, pathogenicity, colony morphology, the ability to respond to a variety of stress conditions, etcetera. They conclude that FlbA and FlbC play an important role in all these processes while the role in asexual development for FlbB, FlbE and FlbD would be minor.

In my opinion, the manuscript is of interest for the readers of Journal of Fungi but there are several points that must be addressed before it is accepted for publication. The first one is the level of the language. The authors should carefully review it and rewrite several sections. I recommend the assistance of a native speaker. I attach a pdf copy of the manuscript with some (only some of them) corrections.

Another major point is that it will be difficult to determine genetic and functional relationship among the genes and proteins analyzed by checking only single null mutants. I understand that maybe the methodology is not ready for the generation of at least double deletion mutants but they are required. This means that the authors should tone down some statements in the manuscript. A third point is that, in my opinion, the link between of FlbD and the response to oxidative stress caused by H2O2 is too speculative. The authors have analyzed the phenotype of the null flbD mutant only at one concentration of H2O2.  Additional concentrations should be analyzed, at least. Finally, the authors should try to tag FlbE using a different strategy. If the one they used does not work, maybe tagging in the N-terminus, or overexpression of the tagged version of FlbE would work (althoug it is true that in A. nidulans, tagging of a wild-type version of FlbE in the N-terminus causes a fluffy phenotype). This analysis is important because the authors suggested that FlbE includes a NLS in its sequence. In A. nidulans, a nuclear localization for FlbE has not been described.

Taking everything into consideration, my decission is major review.

Author Response

Please see attached a file.

Reviewer 2 Report

In this report, Guo et al have characterized the five "fluffy" genes (flbA-flbE), that form a core part of the filamentous fungal conidiation pathway, in the insect pathogenic fungus, B. bassiana via construction and characterization of targeted gene knockouts of each gene. A wide range of phenotypes from growth and development to stress response to virulence were examined for each mutant. These data constitute an important analyses of these genes in this group of fungi and expand our understanding of the contributions of conidiation pathways in a range of physiological processes. The experimental design and presentation is well done and overall the work would be an excellent addition to the field in this area. 

The only criticism is that the language throughout the manuscript requires extensive revision. The first sentence of the abstract is long and confusing, rather than "dispensable", "are not required" would be better, "stressful" is an adjective and not a condition, "more lowered" is grammatically awkward, and the second clause of this sentence makes no sense. -These are only the first three sentences of the abstract. Would strongly recommend use of a professional English language editing service before this paper is acceptable.

Author Response

Please see attached a file.

Reviewer 3 Report

While the subject is interesting, the paper missed clear presentation of novelty, importance and purpose of performed research. Also, more than half of referenced papers are auto-citation. Literature searching seems poor.

Line 5o –‘to date ..’ -authors reference paper from 1995!

Line 51-reference [286]?

Lines 60-73 should be rephrased  - in current form difficult to read

Lines 253 except for its unpredictability ? not clear

Figure 1 Images in Panel C should be marked GFP/DAPI/merge where applicable

Lines262-292 should be rephrased for clarity

Conclusions are weak

Author Response

Please see attached a file.

Round 2

Reviewer 1 Report

I have checked the new version of the manuscript and the response of the authors to the queries of the three reviewers. The authors have done a great effort in rewriting the manuscript but it still needs further english editing (see for example the new version of the abstract). I recommend again the assistance of a native speaker or editing services.

Regarding the response of the authors to my comments to the first version of the manuscript, I still think that additional H2O2 concentrations should be tested with the null flbD mutant. The authors state that they have optimized the concentration of H2O2 used taking the wild-type strain as the reference but the behavior of the mutant could be completely different to that of the wild-type. And if there is sensitivity of the null flbD mutant, then a trend in that sensitivity should be observed at increasing concentrations of H2O2.

Finally, and if there are two selection markers available in this fungus, it is not clear to me why, and at least, a double-null mutant of flbA and flbC has not been generated. I would like to see a comment on this in the manuscript, before acceptance.

Taking everything into consideration, my recommendation is again Major Review

Author Response

I have checked the new version of the manuscript and the response of the authors to the queries of the three reviewers. The authors have done a great effort in rewriting the manuscript but it still needs further english editing (see for example the new version of the abstract). I recommend again the assistance of a native speaker or editing services.

Authore response: Thanks a lot for encouragement.

Regarding the response of the authors to my comments to the first version of the manuscript, I still think that additional H2O2 concentrations should be tested with the null flbD mutant. The authors state that they have optimized the concentration of H2O2 used taking the wild-type strain as the reference but the behavior of the mutant could be completely different to that of the wild-type. And if there is sensitivity of the null flbD mutant, then a trend in that sensitivity should be observed at increasing concentrations of H2O2.

Author response: The suggested experiemnt has been carried out. The new data are shown in added Figure 2D and 2E, well supporting our previous observation.

Finally, and if there are two selection markers available in this fungus, it is not clear to me why, and at least, a double-null mutant of flbA and flbC has not been generated. I would like to see a comment on this in the manuscript, before acceptance.

Author response. Yes, only two herbicide-resistant markers are available for manipulation of tareget genes in B. bassiana and theoretically possible for use in constructing double deletion mutants. However, the success of double deletion based on the two markers is only something lucky.  I have seen a few successful double deletions in B. bassiana in the past decade. By the way, I don't think that the double deletion of flbA and flbC is critical for the purpose of our study. I don't understand a reason to do so. 

Reviewer 2 Report

Language has been improved. Please go over one last time for some minor edits and awkward sentences/grammatical issues.

Author Response

Thanks a lot for understanding and encouragement. I have tried my best to revise the manuscript.